# VB-Net: Voxel-Based Broad Learning Network for 3D Object Classification

**Zishu Liu [1], Wei Song [1,2,*] , Yifei Tian [3], Sumi Ji [4], Yunsick Sung [4] , Long Wen [5], Tao Zhang [6], Liangliang Song [7] and Amanda Gozho [1]**

[1]  School of Information Science and Technology, North China University of Technology, Beijing 100144, China; 2018312120105@mail.ncut.edu.cn (Z.L.); amzyjones811@gmail.com (A.G.)

[2]  Brunel London School, North China University of Technology, Beijing 100144, China

[3]  Department of Computer and Information Science, University of Macau, Macau 999078, China; yb87403@um.edu.mo

[4]  Department of Multimedia Engineering, Dongguk University, Seoul 04620, Korea; ji.si@mme.dongguk.edu (S.J.); sung@dongguk.edu (Y.S.)

[5]  Beijing Municipal Engineering Research Institute, Beijing 100037, China; long.wen@roadwaysmart.com

[6]  Beijing Key Laboratory of Road Engineering Materials and Testing & Inspection Technology, Beijing 100144, China; tao.zhang@roadwaysmart.com

[7]  Roadway Smart (Beijing) Technology Co., Ltd., Beijing 100144, China; liangliang.song@roadwaysmart.com

*   Correspondence: sw@ncut.edu.cn; Tel.: +86-10-8880-1912

**Abstract:** Point clouds have been widely used in three-dimensional (3D) object classification tasks, i.e., people recognition in unmanned ground vehicles. However, the irregular data format of point clouds and the large number of parameters in deep learning networks affect the performance of object classification. This paper develops a 3D object classification system using a broad learning system (BLS) with a feature extractor called VB-Net. First, raw point clouds are voxelized into voxels. Through this step, irregular point clouds are converted into regular voxels which are easily processed by the feature extractor. Then, a pre-trained VoxNet is employed as a feature extractor to extract features from voxels. Finally, those features are used for object classification by the applied BLS. The proposed system is tested on the ModelNet40 dataset and ModelNet10 dataset. The average recognition accuracy was 83.99% and 90.08%, respectively. Compared to deep learning networks, the time consumption of the proposed system is significantly decreased.

**Keywords:** broad learning system; point cloud; 3D object classification

## 1. Introduction

In recent years, point clouds have been researched in various fields, such as autonomous robot systems [1], three-dimensional (3D) face recognition [2], intelligence surveillance [3], and 3D modeling [4]. Unlike images captured by cameras, the point clouds are not likely to be influenced by lighting and color, and they maintain a high resolution. However, when using most existing neural network structures, point clouds are hard to process directly because of their irregularity.

To solve this problem, researchers developed various methods to extract features from raw point clouds for existing deep learning structures, such as spin image (SI) [5], clustered viewpoint feature histogram [6], and view feature histogram [7]. As deep learning is gaining success in many fields, researchers also proposed or employed several deep learning structures for different tasks [8–10], e.g., 3D model retrieval. However, most deep learning structures contain a large number of parameters, which increases the time consumption of the whole system. It is also inconvenient to add new categories into a trained deep learning network or change the structure of a deep learning network, because the

whole network then has to be retrained. These problems make it difficult to apply deep learning methods in fields requiring real-time and massive data processing.

Recently, the broad learning system (BLS) was developed and applied for image classification [11]. Compared to deep learning structures, the time consumption of BLS is decreased significantly, and there is no need to retrain the whole network upon changing the network structure. However, in BLS, it is difficult to directly process irregular point cloud data. Thus, a preprocessing of point cloud data is required to employ BLS in 3D object classification tasks. This paper proposes a 3D object classification system using BLS with a pretrained feature extractor for point cloud data. The pretrained feature extractor intends to change irregular point cloud data into regular features for BLS to process. Another goal of the feature extractor is to find more suitable descriptions for original objects, so as to improve classification accuracy. The proposed system was tested on the ModelNet40 dataset and achieved a recognition accuracy of 83.99%, while the time consumption for training and testing was about 30 s in total, which is far shorter than that of deep learning networks. Experimental results imply that the proposed method is applicable in the field of real-time processing of massive data, i.e., environment perception.

The paper is organized as follows: Section 2 surveys related works in 3D object classification area; Section 3 introduces the proposed system; Section 4 describes the experiments and discusses the results; Section 5 gives the conclusion and the suggestions for future research.

## 2. Related Works

Researchers employ many methods in 3D object recognition, i.e., machine learning-based methods, graph-based methods, and feature-based methods. Feature-based methods are usually employed when using point cloud data for classification tasks.

Generally, there were two kinds of methods mainly used for feature-based object recognition: the global feature-based method and the local feature-based method. When adopting global feature-based methods, a segmentation step is required before classification. Drost et al. [12] developed a global descriptor which consisted of point pair features. By grouping similar point pairs, a voting scheme could work on the local space for the classification job. Their methods performed well under a noise situation. Chen et al. developed the global Fourier histogram (GFH) descriptor [13] that used cylindrical angular coordinates. They defined the cylindrical support region in [13] for calculating the GFH by counting the points in each region. With the help of a global reference frame [14] and Fourier transform, GFH also became rotation-invariant. Although global features performed well in many classification tasks, they failed to distinguish objects with similar shapes [13,15]. Thus, researchers introduced some local features to address this and other problems with global features. Guo et al. proposed Tri-Spin-Image (TriSI) [16] and overcame the weak point of the traditional SI descriptor. Employing principal component analysis and a local reference frame, TriSI was robust to noise and mesh resolution. Yang et al. introduced a descriptor named local feature statistics histogram (LFSH) [17], which exploited histograms by counting local features around the chosen points using one-dimensional (1D) arrays. LFSH worked faster than other local feature descriptors, e.g., SI, yet employing 1D histograms led to some information loss.

Numerous deep learning networks were employed for point cloud object classification works in recent years. Although some cloud architectures consume raw point cloud data directly [18,19], many deep learning networks still require various preprocessing of point clouds. The Multiview 3D network proposed by Chen et al. [20] converted 3D point clouds into 3D bounding boxes for multiview feature fusion. By combining these multiview features, their network enabled fast classification of objects. Klokov et al. proposed Kd-network [21] which did not require rasterization of point clouds before inputting into the network. Exploiting the indexing structure called Kd-tree, Kd-network saved the memory usage and reduced time consumption in the training and testing process. Esteves et al. introduced spherical convolutional neural network [22], which converted 3D models into spherical

representations and took these spherical representations as the inputs of the network. The spherical representations reduced the input size and made the model become rotation-invariant at the same time.

As researchers developed voxel representation for point clouds, they employed convolutional neural networks (CNNs) in 3D object classification tasks. Zhou et al. developed VoxelNet [23] that converted a point cloud into even voxels to encode features. The application of VoxelNet made it possible to employ a region proposal network for object classification in a point cloud. Li et al. proposed field-probing neural networks (FPNN) [24] that employed field-probing filters to extract features from voxels. The field-probing filters were adaptively distributed in 3D space, and their shape was changed to perceive 3D space more effectively. Compared to traditional 3D CNNs, FPNN performed well in 3D object classification tasks with low time consumption. Wang et al. introduced NormalNet [25], which combined normal vectors of object surfaces and voxels of the 3D object for classification, because normal vectors of object surfaces had stronger capability in classification than voxels. To minimize parameters and extract features for 3D vision tasks, they also employed reflection–convolution–concatenation for convolutional layers. However, a limitation of deep learning networks was the computational cost increased as the network became deeper. Introducing time-consuming 3D convolution made it worse. Thus, many researchers applied graphics processing units (GPUs) in deep learning architecture for acceleration.

In recent years, Chen et al. introduced a new neural network structure called the broad learning system [11]. Unlike deep learning networks, BLS did incremental training without retraining the whole network, which meant that changing the structure of BLS was easier than deep learning networks. Introducing the pseudo inverse matrix of the weight matrix, the updating process of BLS was also faster than that of deep learning networks. Thus, BLS significantly reduced the time consumption in both training and testing processes, and it was more flexible than most deep learning networks. This is the reason why this paper applied BLS for object classification.

## 3. VB-Net

This section provides a detailed description of the proposed VB-Net, including the BLS training algorithm, the pretraining of our feature extractor, and the calculation of the final output.

### 3.1. System Overview

The structure of the proposed VB-Net is shown in Figure 1. In this system, raw point clouds are voxelized into $d \times d \times d$ voxels, then input into the pretrained feature extractor to extract features. Those features are mapped to the features in the feature layer ($Z_{1,2,\dots,n}$) of the BLS system using mapping matrices $W_i, i = 1, 2, \dots, n$, and the features in enhancement layer ($H_{1,2,\dots,m}$) are calculated using the feature layer. The nodes in the output layer ($o_{1,2,\dots,k}$) are calculated using $Z_{1,2,\dots,n}$ and $H_{1,2,\dots,m}$. This paper defines the transformation matrices between the feature layer and enhancement layer as $W'_j, j = 1, 2, \dots, m$, and the transformation matrix connects output layer with the feature layer and enhancement layer as $W^*$. This paper also defines the feature layer $Z$ as $Z = \{Z_1, Z_2, \dots, Z_n\}$, enhancement layer $H$ as $H = \{H_1, H_2, \dots, H_m\}$, and output layer $O$ as $O = \{o_1, o_2, \dots, o_k\}$. Variables $n, m,$ and $k$ represent the number of nodes in the feature layer, enhancement layer, and output layer, respectively.

### 3.2. VoxNet Feature Extractor

The first step of the proposed system is converting raw point clouds into $d \times d \times d$ voxels. Assuming there are $N$ points in a point cloud, the point cloud dataset is defined as $P = \{p_1, p_2, \dots, p_i, \dots, p_N\}$, where $p_i$ is the $i$-th point in the point cloud, $I \in [1, N]$. Each $p_i$ contains three coordinates $x, y, z$, denoted by $p_i = (x_i, y_i, z_i)$, $p_i \in P$, and the definition of the voxel space $V = \{v_{l,a,b} = 0 \mid l, a, b \in [1,d], l, a, b \in N^+\}$. Points in $P$ are scaled into the range between 0 and 1 before voxelization. Three equations, $l = x_i \times d, a = d - y_i/d$, are employed to calculate to which voxel point $p_i$ belongs, and then increase the value of $v_{l,a,b}$ by one.

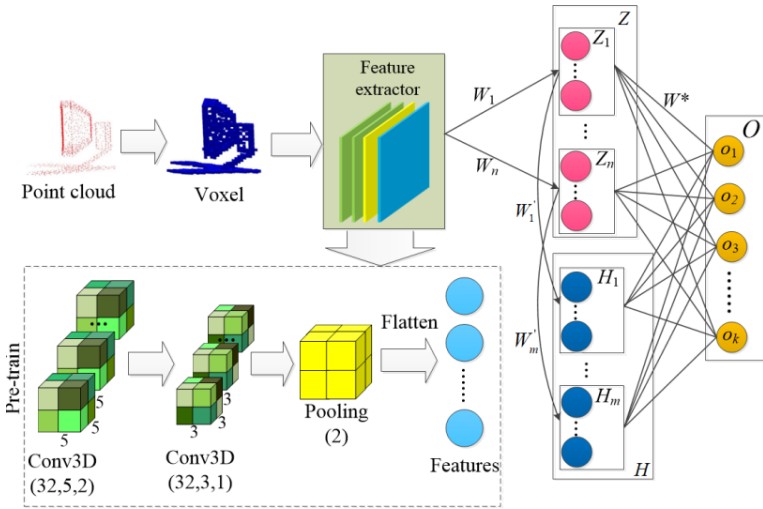

**Figure 1.** The network structure of VB-Net.

We employ a feature extractor that is similar to VoxNet [26] as shown in Figure 1, because the BLS cannot process voxel data directly. Conv3d (*f*, *e*, *s*) represents 3D convolution with *f* filters, kernel size *e*, and stride *s*. Pooling (*p*) means a max pooling layer with pool size *p*. The activation function for Conv3d is ReLU. The outputs of the flatten layer are the features applied by object classification tasks.

The feature extractor also needs a pretraining process to extract more suitable features from input voxel data. In order to obtain better training results, we only keep the last fully connected layer in the original VoxNet [26], as shown in Figure 2. Variable *c* is equal to the number of categories in the dataset. Consequently, the feature extractor is trained as a deep learning network.

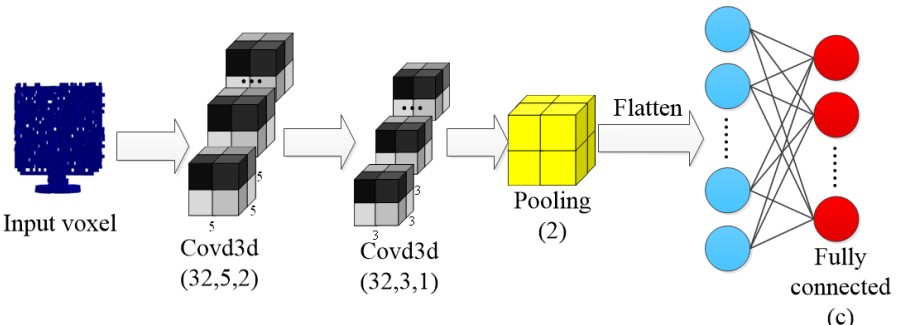

**Figure 2.** Feature extractor architecture for pretraining.

### 3.3. Training of BLS

The features gained by extractor *F* in Section 3.1 are mapped to the feature layer *Z* using Equation (1) where $W_i$ and $\beta_i$ consist of random numbers. To generate nodes in the enhancement layer, Equation (2) is employed where $W'_j$ is a predefined matrix filled with random numbers following the standard normal distribution; $\beta'_j$ is a randomly generalized bias vector in this paper.

$$z_i = \sigma(FW_i + \beta_i) \tag{1}$$

$$h_j = \varphi(ZW'_j + \beta'_j) \tag{2}$$

Nodes in output layer $o_i$, $o_i \in O$, $I \in [1, k]$ are calculated using the following equation in [11] where matrix *W\** is the training target of the training process:

$$O = [Z \,|\, H]W^* \tag{3}$$

This paper defines a new matrix *M* as *M* = [*Z* | *H*]; then, the output matrix *O* is calculated using *MW\**. Therefore, our goal is to narrow the gap between the results of *MW\** and values in output matrix *O*. The ridge regression approximation algorithm is applied for adjusting the values in matrix *W\**. The $l_2$ norm method is adopted for improving the generalization performance of the BLS as shown in Equation (4).

$$W_0^* = \underset{W^*}{\mathrm{argmin}} \|MW^* - O\|_2^2 + \lambda \|W^*\|_2^2 \tag{4}$$

The optimal solution $W_0^*$ is calculated using $W_0^* = M^+ O$, where $M^+$ is the pseudo inverse matrix of matrix *M*. Considering that it is hard to compute $M^+$ directly, an approximation method is adopted here as shown in Equation (5), where matrix *I* is an identity matrix, variable $\lambda \in \mathrm{R}$.

$$W^* = M^+ O = (\lambda I + MM^T)^{-1} M^T O \tag{5}$$

As the output matrix *O* was already computed in Equation (3), only the matrix $M^+$ needs calculation. As mentioned in [11], the solution $M_0^+$ is close to the original pseudo inverse matrix as variable $\lambda$ tends to 0. Thus, Equation (6) is employed to calculate matrix $M^+$.

$$M_0^+ = \lim_{\lambda \to 0} (\lambda I + MM^T)^{-1} M^T \tag{6}$$

The training process of the proposed BLS with the feature extractor is done. If classification accuracy does not meet the required number on the test set, we can increase or decrease the number of nodes in the enhancement layer and/or feature layer to improve the classification accuracy of our model.

## 4. Experiments and Analysis

The proposed VB-Net was tested on the ModelNet40 dataset [27], which contained 12,311 objects from 40 categories, and its orientation-aligned subset ModelNet10 [27]. Samples of point cloud objects in the dataset and its voxel representations are presented in Figure 3. Experiments on ModelNet40 were run on a computer using the Windows 10 operating system, with an Intel® Xeon® (Intel, Santa Clara, CA, USA) Silver 4110 central processing unit (CPU) @ 2.10 GHz, with 16 GB random-access memory (RAM). Experiments on ModelNet10 were run using the Windows 10 operation system, with an Intel® Core™ (Intel, Santa Clara, CA, USA) i7-4720HQ CPU @ 2.59GHz, with 8 GB RAM.

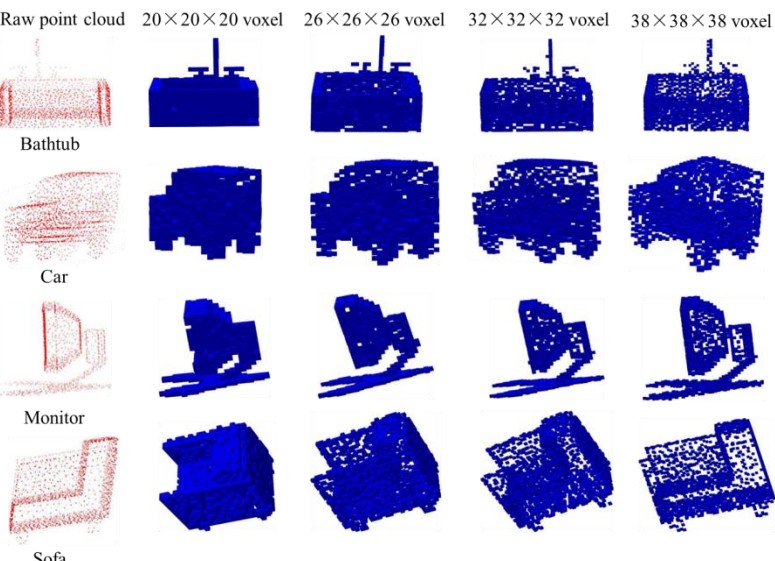

**Figure 3.** Samples in the ModelNet40 dataset.

Point cloud data were converted into voxels under resolutions of $20 \times 20 \times 20$ and $32 \times 32 \times 32$ in our experiments. The feature extractor trained 120 epochs on the ModelNet40 training set as the pretraining process.

### 4.1. Performance of VB-Net

The performance of the BLS is affected by the number of enhancement nodes and the number of feature nodes [11]. Therefore, we designed two sets of experiments to test our system. First, we set 900 feature nodes in our system, increasing the enhancement nodes. We added $26 \times 26 \times 26$ and $38 \times 38 \times 38$ voxel resolutions to observe the performance of VB-Net under different resolutions. Experimental results on the ModelNet40 dataset are shown in Table 1.

**Table 1.** Accuracy under different resolutions and enhancement nodes in VB-Net.

| Enhancement Nodes | Average Accuracy ($20 \times 20 \times 20$ Voxel) | Average Accuracy ($26 \times 26 \times 26$ Voxel) | Average Accuracy ($32 \times 32 \times 32$ Voxel) | Average Accuracy ($38 \times 38 \times 38$ Voxel) |
|---|---|---|---|---|
| 100 | 72.20% | 72.28% | 73.86% | 73.37% |
| 200 | 72.44% | 72.48% | 74.43% | 73.50% |
| 300 | 72.77% | 72.97% | 74.87% | 73.86% |
| 400 | 72.89% | 72.85% | 74.91% | 74.10% |
| 500 | 73.17% | 73.09% | 75.24% | 74.27% |
| 1000 | 81.11% | 80.14% | 80.75% | 79.82% |
| 1500 | 81.24% | 81.11% | 81.07% | 80.22% |
| 2000 | 82.57% | 80.59% | 81.44% | 80.63% |
| 2500 | 82.61% | 80.51% | 81.64% | 80.75% |
| 3000 | 82.49% | 79.57% | 82.41% | 80.38% |
| 3500 | 81.52% | 79.29% | 82.41% | 80.18% |
| 4000 | 81.64% | 79.05% | 80.91% | 79.82% |
| 4500 | 80.42% | 77.87% | 80.02% | 78.64% |
| 5000 | 79.90% | 76.17% | 79.94% | 78.68% |

As shown in Table 1, the average accuracy initially increased with the number of enhancement nodes. The accuracy reached its maximum with 2500 enhancement nodes in VB-Net under $20 \times 20 \times 20$ resolution (82.61%) and $38 \times 38 \times 38$ resolution (80.75%). Under $26 \times 26 \times 26$ resolution or $32 \times 32 \times 32$ resolution, the accuracy reached its max value with 1500 nodes in the enhancement layer (80.75%) or 3000 enhancement nodes in the system (82.41%). A further increase in the number of enhancement nodes no longer improved accuracy; on the contrary, the classification accuracy then decreased with the number of enhancement nodes.

Another set of experiments involved increasing feature nodes in VB-Net. According to Table 1, VB-Net showed good performance with 3000 enhancement nodes in the enhancement layer at both resolutions. Hence, we set the number of enhancement nodes to 3000 and kept increasing the feature nodes in VB-Net. The results of these experiments on the ModelNet40 dataset are shown in Table 2.

The highest accuracy reached 82.41% when 800 feature nodes were used in the proposed network under $20 \times 20 \times 20$ resolution or 80.59% with 400 feature nodes under $26 \times 26 \times 26$ resolution. When 100 feature nodes were used in VB-Net, the highest accuracy reached 83.99% under $32 \times 32 \times 32$ resolution. For $38 \times 38 \times 38$ resolution, the highest accuracy was 80.99% with 500 feature nodes in the system. Unlike the first set of experiments, classification accuracy under $32 \times 32 \times 32$ resolution decreased directly. From the data in Tables 1 and 2, it is evident that the performance of VB-Net improved as more feature nodes or enhancement nodes were utilized in the calculation of final results. When the number of nodes exceeded the peak amount at which performance was best, the performance of VB-Net became worse due to the interference of too much data. The appropriate number varied with voxel resolutions and strategies.

The time consumption in training and test processes for two different training strategies mentioned above under $20 \times 20 \times 20$ voxel resolution was also computed. Although the time consumption increased

with the number of nodes in the network, time consumption with increasing feature nodes increased more quickly than with increasing enhancement nodes, as shown in Figure 4a,b. However, the former strategy held fewer nodes in the network, which was different from the case where time consumption increased with the number of nodes in the network. We tested the situation with 5900 nodes in network (900 feature nodes and 5000 enhancement nodes), where the time consumption was 38.86 s in total, which was far shorter than the method with increasing feature nodes. When we increased the number of feature nodes in BLS, we increased time consumption upon feature mapping, with simultaneous enhancement node generalization and calculation of output, while increasing only the number of enhancement nodes increased the time consumption of the last two steps. This phenomenon is explained in Figure 4.

**Table 2.** Accuracy under different resolutions and feature nodes in VB-Net.

| Feature Nodes | Average Accuracy (20 × 20 × 20 Voxel) | Average Accuracy (26 × 26 × 26 Voxel) | Average Accuracy (32 × 32 × 32 Voxel) | Average Accuracy (38 × 38 × 38 Voxel) |
|---|---|---|---|---|
| 100 | 81.80% | 79.45% | 83.99% | 80.10% |
| 200 | 81.88% | 80.71% | 83.99% | 80.42% |
| 300 | 81.88% | 80.14% | 83.83% | 80.67% |
| 400 | 81.40% | 80.59% | 82.53% | 80.51% |
| 500 | 81.11% | 80.06% | 82.33% | 80.99% |
| 600 | 81.92% | 79.78% | 82.17% | 80.18% |
| 700 | 81.19% | 79.17% | 82.49% | 80.71% |
| 800 | 82.41% | 80.02% | 82.13% | 80.71% |
| 900 | 81.68% | 79.70% | 82.41% | 81.19% |
| 1000 | 81.44% | 79.61% | 81.52% | 79.33% |
| 1500 | 81.40% | 78.80% | 81.03% | 79.74% |
| 2000 | 81.72% | 78.28% | 80.99% | 78.89% |

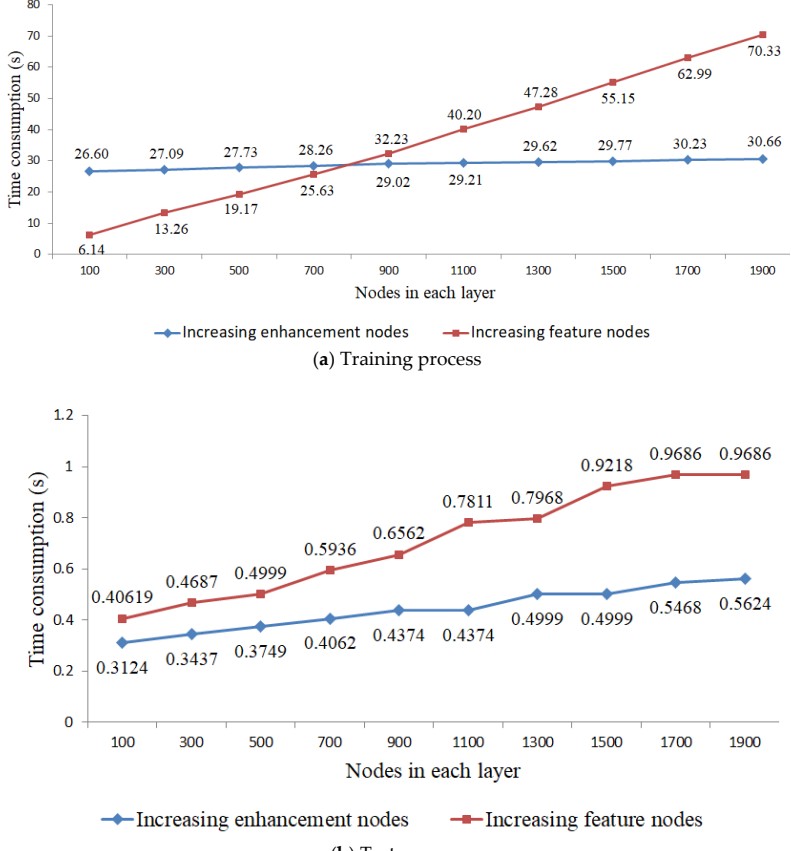

(a) Training process

(b) Test process

**Figure 4.** Relationship between time cost and incremental nodes in VB-Net.

To prove the proposed system had better performance in object classification tasks, we compared our classification results with some existing deep learning networks using the same dataset, as shown in Table 3. Although VoxNet [26] performed better than our method on the ModelNet10 dataset, our method ran faster than VoxNet [26]. We tested the time consumption of VB-Net and VoxNet [26] on ModelNet10, where the training epochs for VoxNet and the feature extractor were 30. The experiment showed that our method took only 39.13 s while VoxNet took 3573.78 s.

**Table 3.** Comparison with existing deep learning networks.

| Methods | Input | Average Accuracy (ModelNet40) | Average Accuracy (ModelNet10) |
|---|---|---|---|
| Our method ($20 \times 20 \times 20$ voxel) | Voxel | 82.41% | 89.64% |
| Our method ($32 \times 32 \times 32$ voxel) | Voxel | 83.99% | 90.08% |
| 3DShapeNets [27] | Voxel | 77% | 83.5% |
| VoxNet [26] | Voxel | 83% | 92% |
| DeepPano [28] | Image | 77.63% | 85.45% |
| Geometry Image [29] | Image | 83.9% | 88.4% |
| Soltani et al. [30] | Image | 82.10% | - |
| ECC [31] | Graph | 83.2% | 90.0% |
| PointNet [32] | Point | - | 77.6% |

## 4.2. Performance of Feature Extractor

To evaluate the effect of our feature extractor, we employed an ablation experiment on the ModelNet10 dataset, as shown in Table 4, with $32 \times 32 \times 32$ voxel resolution. We set 450 feature nodes and 1100 enhancement nodes in BLS and VB-Net. The feature extractor was trained 30 times. FE refers to the VoxNet feature extractor and OFE refers to the original VoxNet [26]. The "time" column in Table 4 records the total time consumption (training time pluses test time) of one module.

**Table 4.** Ablation experiment of VB-Net. FE, feature extractor; OFE, original feature extractor (VoxNet).

| Model | FE | BLS | OFE | Average Accuracy | Time (s) |
|---|---|---|---|---|---|
| A | | √ | | 86.78% | 129.36 |
| B | √ | | | 85.35% | 3111.12 |
| C | √ | √ | | 89.86% | 39.13 |
| D | √ | | √ | 89.75% | 38.40 |
| E | | | √ | 86.12% | 3573.78 |

Comparing models A, C, and D, the feature extractor had great influence on total time consumption. BLS with a feature extractor ran much faster than the original BLS, and the classification accuracy also improved by about 3%. According to average accuracy and total time consumption, BLS performed better than the deep learning network. Although model E reached higher classification accuracy than model B, model C still achieved higher accuracy than model D. This proves that our feature extractor is more efficient than VoxNet [26] when combined with BLS.

We adopted another ablation experiment to examine whether our feature extractor could be more efficient. Experiments were done on the ModelNet10 dataset under $20 \times 20 \times 20$ voxel resolution using feature extractors trained 120 times. Settings of VB-Net were 420 feature nodes and 900 enhancement nodes. Results are presented in Table 5.

Model F gained higher accuracy than our method (model I) because more features were input into BLS in model F, but our method ran faster than F. When comparing models F, G, and H, the max pooling layer had an influence on the performance of the feature extractor. Without the max pooling layer, the classification accuracy of VB-Net dropped about 2%. Performance on models G and F proved that the first 3D convolutional layer not only reduced the input features of BLS, but also improved the accuracy. The second convolutional layer helped VB-Net run more efficiently by reducing the input features of BLS.

**Table 5.** Ablation experiment of feature extractor.

| Model | Conv3D (32,5,2) | Conv3D (32,3,1) | Max Pooling | Accuracy | Training Time (s) | Test Time (s) |
|---|---|---|---|---|---|---|
| F | √ | | √ | 89.86% | 2.91 | 0.1149 |
| G | | √ | √ | 89.42% | 44.78 | 0.4047 |
| H | √ | √ | | 87.77% | 13.72 | 0.1928 |
| I | √ | √ | √ | 89.64% | 1.91 | 0.0829 |

### 4.3. Noise Resistance Test

To examine noise resistance in our model, we added Gaussian noise to original point clouds, and then generated the voxel. Figure 5 gives examples under Gaussian noise with $20 \times 20 \times 20$ and $32 \times 32 \times 32$ voxel resolution with original voxels in the ModelNet40 and ModelNet10 datasets. We employed our method, VoxNet [26], and model D mentioned in Section 4.2 in the experiments. Experimental results are shown in Table 6. The VoxNet [26], feature extractors in model D, and VB-Net were trained 120 times on the training set (accuracies of VoxNet [26] on the ModelNet40 and ModelNet10 datasets under $32 \times 32 \times 32$ voxel resolution were cited from [26] directly).

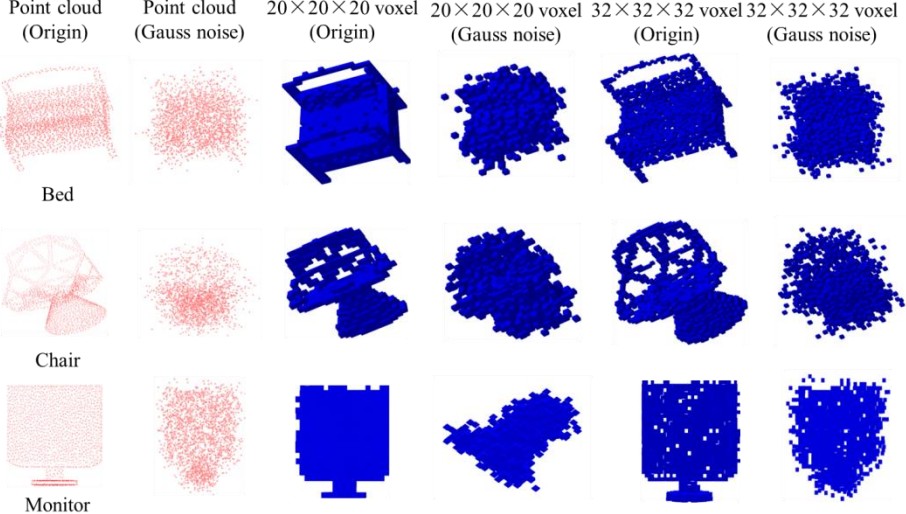

**Figure 5.** Samples with Gaussian noise and their original shape.

VoxNet [26] was the most affected by Gaussian noise, and its accuracy dropped by half on ModelNet10, with even worse performance on ModelNet40. The accuracies of methods with BLS decreased much more slightly compared to VoxNet [26], while some even increased a little. When comparing the original BLS and the BLS employing VoxNet [26] as the feature extractor, our method also had better performance. We carried out more experiments with different scales of Gaussian noise to test the robustness of our system, as shown in Figure 6.

**Table 6.** Classification accuracies of original data and Gaussian noise data.

| Method | ModelNet10 | | | | ModelNet40 | | | |
|---|---|---|---|---|---|---|---|---|
| | Accuracy (32 × 32 × 32) | Accuracy (32 × 32 × 32) (No Noise) | Accuracy (20 × 20 × 20) | Accuracy (20 × 20 × 20) (No Noise) | Accuracy (32 × 32 × 32) | Accuracy (32 × 32 × 32) (No Noise) | Accuracy (20 × 20 × 20) | Accuracy (20 × 20 × 20) (No Noise) |
| BLS | 83.92% | 86.01% | 87.55% | 86.56% | 72.33% | 76.86% | 77.99% | 79.13% |
| Our method | 90.63% | 90.08% | 90.08% | 89.64% | 82.78% | 83.99% | 81.24% | 82.41% |
| Model D | 89.53% | 89.75% | 89.31% | 88.76% | 82.05% | 81.07% | 81.56% | 80.87% |
| VoxNet [26] | 42.73% | 92% | 47.79% | 87.88% | 17.26% | 83% | 20.98% | 76.29% |

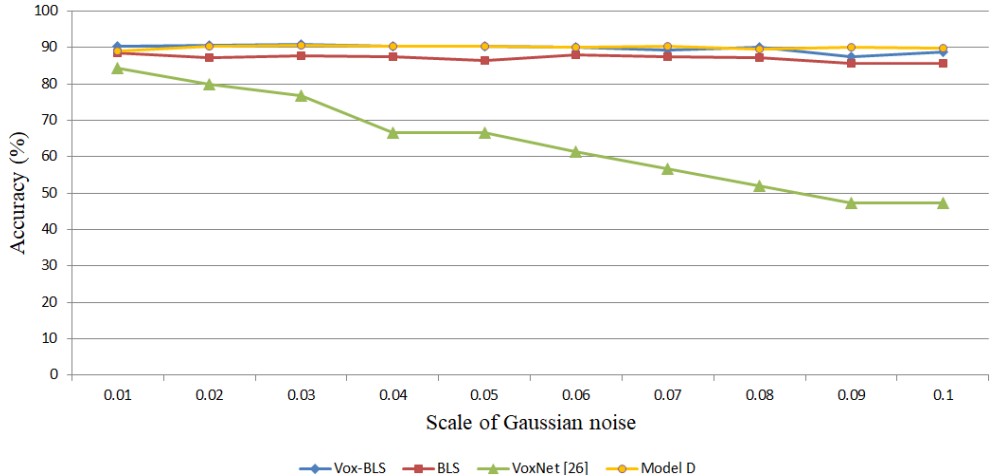

**Figure 6.** Accuracy under different scales of Gaussian noise.

Obviously, the accuracy of VoxNet [26] decreased as the scale of noise increased, while the accuracies of methods with BLS were stable at about 90%. We also noticed that, in some experiments, the accuracy of data with noise was higher than that without noise. This situation happened in all methods with BLS. We think this phenomenon was mainly due to the settings in BLS. According to Section 4.1, the performance of BLS was influenced by the number of nodes in the feature layer and enhancement layer. The reason for the improvement in accuracy was that we did not find the best combination of enhancement nodes and feature nodes in the experiments. Considering that there is no theory regarding the relationship among accuracy, nodes in BLS, and voxel resolutions, and considering that the number of possible combinations is large, we think that meeting this phenomenon is acceptable.

## 5. Conclusions

This paper developed a 3D object classification system using VB-Net for point cloud data. The proposed system employed voxels to present point clouds, whereas a pretrained VoxNet was employed as a feature extractor, and the BLS was used for classification tasks. The proposed system was tested on the ModelNet40 dataset and ModelNet10 dataset. The average accuracies were as follows: 83.99% under voxel resolution of $32 \times 32 \times 32$ and 90.08% under voxel resolution of $20 \times 20 \times 20$ on ModelNet40; 82.41% under voxel resolution of $32 \times 32 \times 32$ and 89.64% under voxel resolution of $20 \times 20 \times 20$ on ModelNet10. An appropriate increment in voxel resolution improved the performance of the proposed system. Equally, an appropriate addition of enhancement nodes and/or feature nodes improved the classification accuracy of VB-Net. The proposed VB-Net also showed resistance in point cloud data with Gaussian noise. In future work, we will try to find compositions of enhancement nodes and feature nodes in VB-Net to achieve higher classification accuracy.

**Author Contributions:** Methodology, Z.L.; project administration, W.S.; supervision, Y.T. and L.W.; funding acquisition, Y.S., L.S., and T.Z.; investigation, S.J. and A.G. All authors have read and agreed to the published version of the manuscript.

**Funding:** This research was fund by the MSIT (Ministry of Science, ICT), Korea, under the High-Potential Individuals Global Training Program (2020-0-01576) supervised by the IITP (Institute for Information and Communications Technology Planning and Evaluation), the National Nature Science Foundation of China (No. 61503005), the Great Wall Scholar Program (CIT&TCD20190304, CIT&TCD20190305), the National Key R&D Program of China under Grant (2017YFC0821102, 2917YFC0822504), and the "Yuyou" and Education Reform Projects of North China University of Technology.

**Conflicts of Interest:** The authors declare no conflict of interest.

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
