# Peer review of "VB-Net: Voxel-Based Broad Learning Network for 3D Object Classification"

_applsci, doi:10.3390/app10196735_

Round 1

Reviewer 1 Report

The paper presents a new methodology for 3D point cloud classification. The topic is interesting but some corrections are needed:

  1. references are a bit "old". can the author search for papers more recent or explain why there is no new bibliography about the topc?
  2. the 3D point clouds analysed refer to objects with an easy geometry. Did the authors tested it on more complex objects? if no, have they the intention to do it? It can be interesting for the application in different fields of research.
  3. Even if the procedure is explained properly, maybe a deepen analysis on the obtained results is needed, not only tables regarding accuracy (that are important).

Reviewer 2 Report

This paper developed a 3D object classification system using VB-Net for point cloud data. The proposed method employes to "voxel clouds" a pre-trained VoxNet as feature extractor, and the BLS for classification tasks. The proposed system was tested and positively verified for high classification accuracy. I recommend publication of the paper with minor language amendments.
